# Characterization of Wei Safflower Seed Oil Using Cold-Pressing and Solvent Extraction

**DOI:** 10.3390/foods12173228

**Published:** 2023-08-28

**Authors:** Linlin Song, Sheng Geng, Benguo Liu

**Affiliations:** 1College of Life Science and Technology, Henan Institute of Science and Technology, Xinxiang 453003, China; cnusll@126.com; 2School of Food Science, Henan Institute of Science and Technology, Xinxiang 453003, China; gengshenggs@126.com

**Keywords:** Wei safflower seed oil, cold pressing extraction, physicochemical properties, oil characterization

## Abstract

Wei safflower seed oil (WSO) prepared by the cold pressing method and organic solvent extraction method was characterized in this study. The yield of cold-pressed WSO (CP-WSO) was inferior to that of *n*-hexane-extracted WSO (HE-WSO). The physicochemical properties (refractive index, density, iodine value, insoluble impurities) and fatty acid compositions were similar, and both were rich in linoleic acid. However, CP-WSO had better color and less solvent residue. The type and content of vitamin E in CP-WSO was also superior to that in HE-WSO, which explained the high oxidative stability of CP-WSO in the Rancimat test. Our results provide a reference for the development of Wei safflower seed oil.

## 1. Introduction

Safflower (*Carthamus tinctorius* L.) is a common medicinal plant belonging to the genus *Carthamus* in the Asteraceae family [1]. It is mainly distributed in South America, North America and Asia [2]. The planting area of safflower in China is about 33,300 hm^2^. In Weihui, Yanjin and Fengqiu counties of Xinxiang city, China, a safflower variety with regional characteristics is called Wei safflower. In May 2010, Wei safflower was successfully declared the “Geographical Indication of Agricultural Products of the People’s Republic of China”. It is well-known throughout the country and enjoys a high reputation in the international market. Safflower oil is made from safflower seeds and is widely used in food, medicine and other fields because it is rich in linoleic acid [3,4].

The traditional preparation techniques of vegetable oils include heat-pressing extraction and organic solvent extraction [5]. The crude oil produced by the heat-pressing method is not only of low quality, but also has serious protein denaturation of seed meal, which increases the difficulty of subsequent refining [6]. Organic solvent extraction has the advantages of high oil yield and low cost. However, the crude oil produced by organic solvent extraction has safety hazards such as solvent residues [7], and the residual solvents cause damage to human liver and other organs. Supercritical CO_2_ fluid extraction can effectively solve the problem of solvent residue in oil. However, its equipment investment is relatively large, so it is not suitable for the current edible oil processing industry. Oil production by cold pressing refers to the process of destroying the plants’ oil cells by mechanical procedures without the application of heat, allowing oil to flow out from the raw materials [8]. The cold pressing method is a physical method which pressurizes without heating and has no effect on oil and active components. In addition to the general characteristics of the ordinary oil production process, this process can also improve the quality of oil, avoid harmful substances such as trans fatty acids and oil polymers caused by high-temperature processing, and retain the active substances in the oil. It can also avoid the problem of residues of harmful substances such as acid, alkali and heavy metals caused by chemical additives in the refining process, shorten the processing process and save on production cost. Moreover, the nutritional value of pressed cake is improved; protein, dietary fiber and other nutritional components are not denatured; and active substances are preserved, ensuring the development and utilization value of cake. Therefore, this technology is suitable for simultaneous production of high-quality oils and macromolecular nutrients from oil crops [7,8,9]. At room temperature, the fluidity of oil is poor, so the oil yield of cold pressing method is low, but various natural nutrients of oil are preserved in the production. The freshly cold-pressed linseed oil is widely recognized as one of the most popular functional foods for the high content of heat-sensitive omega-3 α-linolenic acid [10].

In recent years, as people’s attention to food safety has increased year by year, people are more fond of natural green food, so research on the cold pressing method to produce various natural vegetable oils is gradually increasing [11,12]. However, there is no report on the preparation of Wei safflower seed oil (WSO) by cold pressing. In view of this, this study adopted cold pressing method and organic solvent extraction method to prepare WSO, and systematically compared the oil quality. The obtained results can prompt the development of Wei safflower seed oil.

## 2. Materials and Methods

### 2.1. Materials and Chemicals

The mature Wei safflower seeds (variety BH-1) were harvested in June 2022 and provided by Xinfulin planting cooperative (Xinxiang, China). Fatty acid methyl esters and tocopherols were the products of Sigma-Aldrich (St. Louis, MO, USA). All other chemicals were of analytical grade.

### 2.2. Chemical Composition Determination

The moisture content of Wei safflower seeds was determined by the oven-drying method (AOAC 925.10). The crude lipid content was measured based on the Soxhlet extraction method (AOAC 923.05). The crude protein content was determined using a Kjeltec analyzer (AOAC 992.23). The crude fiber content was obtained on a Foss M6 fibertec analyzer (AOAC 978.10).

### 2.3. Preparation of WSO

The fresh crushed seeds (1000 g) were pressed with a 6YZ180 automatic hydraulic press (Zhengzhou Bafang Machinery and Equipment Co., Ltd., Zhengzhou, China) at 25 °C and 50 MPa for 0.5 h to obtain cold pressed WSO (CP-WSO). The weight of CP-WSO was recorded to calculate the oil yield.

The fresh crushed seeds (1000 g) were soaked in 10 times volume of *n*-hexane at 25 °C for 3 h, filtered and then evaporated to remove the solvent to obtain hexane-extracted WSO (HE-WSO). Its weight was also recorded to calculate the oil yield of HE-WSO.

### 2.4. Determination of Physicochemical Properties of WSO

The refractive index, density, acid value, iodine value, saponification value, color, moisture and volatile content of WSO were determined according to the AOCS Official Method: Cc 7-25, Ea 7-95, Cd 3d-63, Cd 1d-92, Tl 1a-64, Cc 13e-92 and Ca 2c-25, respectively.

### 2.5. Fatty Acid Composition Determination of WSO

The fatty acid composition of WSO was determined according to a previous report [13]. The analysis was performed on an Agilent 7890A gas chromatography (Santa Clara, CA, USA) with an FID detector and an HP-88 capillary column (100 m × 0.25 mm, 0.2 μm). The injector and detector temperatures were 240 and 280 °C, respectively. Column temperature program was 140 °C (5 min) isotherm, then increased to 240 °C at the rate of 4 °C/min and was held at 240 °C for 10 min. N_2_ was used as the carrier gas with the flow rate of 22 mL/min and a split ratio of 1:50. The injection volume was 1 μL.

### 2.6. Vitamin E Measurement of WSO

The vitamin E measurement of WSO was determined based on a published method using a Waters e2695 high performance liquid chromatography with a fluorescence detector [14]. The separation was carried out on an Elite amino column (250 mm × 4.6 mm, 5 μm) at 40 °C. The mobile phase consisted of *n*-hexane and isopropanol (87:13, *v/v*). The flow rate was kept at 0.8 mL/min. The excitation and emission wavelengths were 298 nm and 325 nm respectively.

### 2.7. Oxidative Stability Evaluation of WSO

The oxidative stability evaluation of WSO was evaluated using a 743 Rancimat analyzer (Metrohm, Herisau, Switzerland) [15]. The sample usage was 3.0 g, the test temperature was set at 110 °C, and the air flow rate was 20 L/h. The induction time was automatically determined by the Rancimat analyzer.

### 2.8. Statistical Analysis

All data were measured in parallel 3 times and expressed as mean ± standard deviation. The statistical comparison was based on Duncan’s test with a confidence level of 95%.

## 3. Results and Discussion

### 3.1. Composition of Wei Safflower Seeds

Wei safflower seed is smaller than sunflower seed. It is oval, white and four-angled. The outer layer is a relatively thick white protective shell (Figure 1). Some of the shells are smooth and some are ridged. A thin seed coat is attached to the shell, and the embryo is wrapped in the seed coat. The composition analysis of Wei safflower seeds is shown in Table 1. Because of its thick protective shell, its fiber content was as high as 57.90%. Its lipid and protein contents were 19.25% and 14.79%, respectively, which was of great development value.

### 3.2. Physicochemical Properties of WSO

The physicochemical properties of CP-WSO and HE-WSO are shown in Table 2. Cold pressing avoids the adverse effects of traditional heat pressing and solves the problem of residual organic solvents, but its oil yield is low. In this experiment, the yield of CP-WSO (9.26%) was significantly inferior to that of HE-WSO (16.42%). There was no significant difference in refractive index, density, iodine value and insoluble impurities between CP-WSO and HE-WSO. However, the moisture and volatile substances of HE-WSO were significantly higher than that of CP-WSO, which could be due to the residue of the solvent used in the extraction process. A significant difference in color between CP-WSO and HE-WSO was also observed, and the color of CP-WSO was more red, which was consistent with the previous conclusion that cold pressed oil can retain the natural color of oil [16].

### 3.3. Fatty Acid Composition of WSO

The fatty acid compositions of CP-WSO and HE-WSO are exhibited in Table 3. The fatty acid compositions of CP-WSO and HE-WSO were similar. Both were rich in unsaturated fatty acids such as linoleic acid (C18:2) and oleic acid (C18:1), while the saturated fatty acids were mainly palmitic acid, which was low in content. In the report of Al Juhaimi et al. [17], oleic acid and linoleic acid were the main fatty acids of safflower oil, and their contents were 41 and 49%, respectively [14]. Erol et al. and Matthaus et al. found the major fatty acid of safflower seed oil was linoleic acid with a mean value of >70% [18,19], which coincided with our results. These differences could be due to different varieties and growing environments. Linoleic acid has high nutritional value, which can prevent the deposition of serum cholesterol in the blood vessel wall, and cure atherosclerosis and cardiovascular diseases [20], so WSO has a potential health-promoting function. 

### 3.4. Vitamin E Composition of WSO

Vitamin E is a fat-soluble vitamin with eight types, which has outstanding biological activities such as antioxidation, cardio-cerebrovascular maintenance, anti-tumor and so on [21]. As shown in Table 4, α-tocopherol was the main type of vitamin E in WSO, which was consistent with the previous report [17]. The content of vitamin E in CP-WSO was significantly higher than that in HE-WSO, and four types of vitamin E were detected in CP-WSO, but only two were detected in HE-WSO. The cold-pressing method destroys the plant tissue through mechanical extrusion as it extracts the oil from the sample, and the composition of the oil is close to the natural state. The solvent method mainly extracts oil from plants based on the similar compatibility principle, which has selectivity in the extraction process. The contents of β-tocopherol and β-tocotrienol in safflower seeds were very low, whereas a large amount of α-tocopherol existed. The *n*-hexane could not have a good dissolution effect on β-tocopherol and β-tocotrienol, so their contents in the extracted oil were too low, which was not conducive to detection.

### 3.5. Oxidative Stability of WSO

Oxidative stability is the sensitivity to characterize the automatic oxidation deterioration of oil, that is, the ability of oil to resist automatic oxidation, which reflects the storability of oil. Oil rich in unsaturated fatty acids is easy to be oxidized, and its oxidation stability is closely related to its antioxidant components. The Rancimat method is a modern method to evaluate the oxidative stability of oils. It can reflect not only the oxidation degree of oil, but also the activity of antioxidant components in oil. The whole test is standardized and automatic, and the oxidative stability of oil can be evaluated by measuring the induction time of oils [22]. Long induction time means good stability. In Figure 2 and Table 5, the induction time of CP-WSO was significantly higher than that of HE-WSO, indicating that the stability of CP-WSO was obviously superior to that of HE-WSO, which may be attributed to the higher vitamin E content of HE-WSO.

## 4. Conclusions

Wei safflower seeds were rich in protein and oil. Compared with HE-WSO, the yield of CP-WSO was lower. However, their physicochemical properties and fatty acid composition were similar, and both were rich in linoleic acid. The type and content of vitamin E in CP-WSO were superior to that in HE-WSO, so the oxidative stability of oil was also better than that of HE-WSO. In general, the quality of CP-WSO is better. Our results indicate that the cold-pressing method can be used to produce high-quality Wei safflower seed oil, but in order to achieve its large-scale production, the optimization of cold pressing parameters or the adoption of multiple cold-pressing treatment is needed to improve its yield.

## Figures and Tables

**Figure 1 foods-12-03228-f001:**
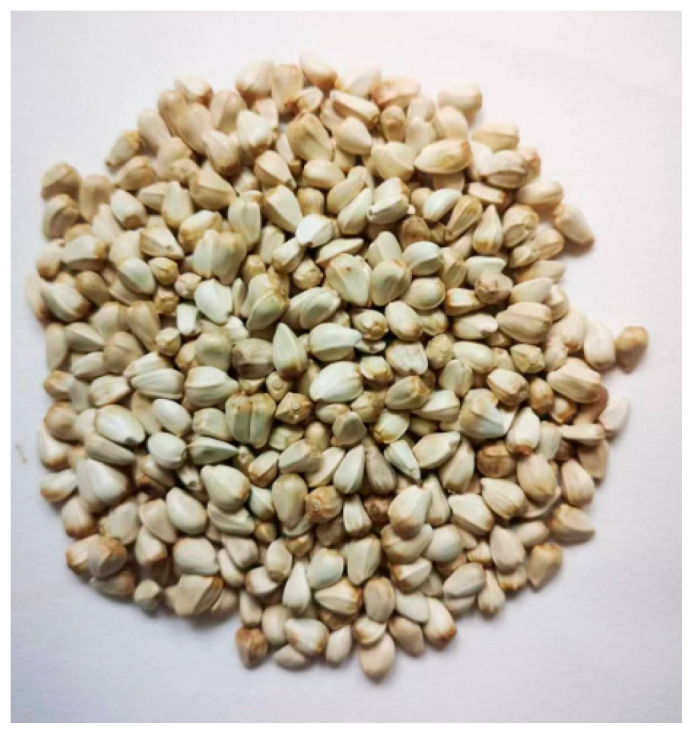
Appearance of Wei safflower seeds.

**Figure 2 foods-12-03228-f002:**
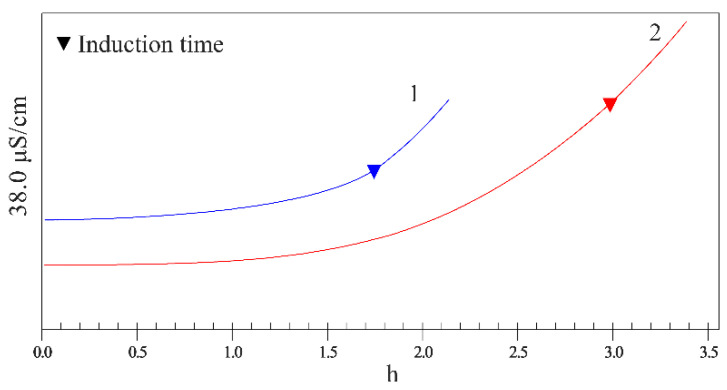
Oxidative stability of *n*-hexane-extracted Wei safflower seed oil (HE-WSO, 1) and cold pressed Wei safflower seed oil (CP-WSO, 2) in Rancimat test.

**Table 1 foods-12-03228-t001:** Composition of Wei safflower seeds.

	Moisture	Lipid	Protein	Fiber	Mineral
Content (%)	6.04 ± 0.00	19.25 ± 0.02	14.79 ± 1.43	57.90 ± 2.64	2.31 ± 0.01

**Table 2 foods-12-03228-t002:** Physicochemical properties of cold pressed Wei safflower seed oil (CP-WSO) and *n*-hexane-extracted Wei safflower seed oil (HE-WSO).

	HE-WSO	CP-WSO
Oil yield (%, *w/w*)	16.42 ± 0.00 ^a^	9.26 ± 0.02 ^b^
Density (g/mL)	0.9247 ± 0.0004 ^a^	0.9254 ± 0.0003 ^a^
Refractive index (n20/D)	1.4732 ± 0.0002 ^a^	1.4726 ± 0.0004 ^a^
Iodine value (g/100 g)	128.51 ± 0.85 ^a^	129.70 ± 2.04 ^a^
Saponification value (mg KOH/g)	177.23 ± 2.47 ^b^	184.67 ± 3.43 ^a^
Acid value (mg KOH/g)	11.65 ± 0.22 ^a^	9.70 ± 0.57 ^b^
Insoluble impurities (%, *w/w*)	0.27 ± 0.06 ^a^	0.29 ± 0.08 ^a^
Moisture and volatile substances (%, *w/w*)	4.39 ± 0.23 ^a^	0.13 ± 0.02 ^b^
Color (Lovibond, 1 in.)	Y 53 ± 2.83 ^a^, R 2 ± 0.00 ^b^	Y 55 ± 2.83 ^a^, R 2.9 ± 0.00 ^a^

Different lowercase superscript letters in the same row indicate significant differences (*p* < 0.05).

**Table 3 foods-12-03228-t003:** Fatty acid composition of cold pressed Wei safflower seed oil (CP-WSO) and *n*-hexane-extracted Wei safflower seed oil (HE-WSO) (%).

	HE-WSO	CP-WSO
C16:0	6.59 ± 0.05 ^a^	6.65 ± 0.05 ^a^
C18:0	1.61 ± 0.01 ^b^	1.72 ± 0.02 ^a^
C18:1	14.20 ± 0.09 ^a^	14.38 ± 0.14 ^a^
C18:2	74.94 ± 0.41 ^a^	74.01 ± 0.56 ^a^
C20:0	0.81 ± 0.01 ^a^	0.47 ± 0.01 ^b^
C20:1	2.26 ± 0.15 ^b^	3.02 ± 0.53 ^a^

Different lowercase superscript letters in the same row indicate significant differences (*p* < 0.05).

**Table 4 foods-12-03228-t004:** Tocopherol contents of cold pressed Wei safflower seed oil (CP-WSO) and *n*-hexane-extracted Wei safflower seed oil (HE-WSO) (mg/100 g oil).

	HE-WSO	CP-WSO
α-Tocopherol	210.15 ± 3.91 ^a^	193.83 ± 3.74 ^b^
β-Tocopherol	—	21.93 ± 0.79
γ-Tocopherol	4.84 ± 0.41 ^a^	5.21 ± 0.42 ^a^
β-Tocotrienol	—	6.52 ± 0.60
Total	214.99 ± 4.31 ^b^	227.49 ± 1.97 ^a^

Different lowercase superscript letters in the same row indicate significant differences (*p* < 0.05).

**Table 5 foods-12-03228-t005:** Induction times of cold pressed Wei safflower seed oil (CP-WSO) and *n*-hexane-extracted Wei safflower seed oil (HE-WSO) (h).

	HE-WSO	CP-WSO
Induction time (h)	1.76 ± 0.03 ^b^	2.97 ± 0.10 ^a^

Different lowercase superscript letters in the same row indicate significant differences (*p* < 0.05).

## Data Availability

The data that support the findings of this study are available on request from the corresponding author. The data are not publicly available due to privacy or ethical restrictions.

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
