# Peer review of "Characterization of Wei Safflower Seed Oil Using Cold-Pressing and Solvent Extraction"

_foods, 2023, doi:10.3390/foods12173228_

Round 1

Reviewer 1 Report

The publication Characterization of Cold-pressed Wei Safflower Seed Oil is interesting to some extent and could bring new information regarding the possibility of using cold-pressing as a method of obtaining safflower oil. However, the authors gave a very cursory treatment of the subject from the introduction to the paper, through the planning of the experiment to the discussion of their own results. The paper is short, and full of gaps. There are no statistical tests to describe the results, no discussion, and no in-depth literature analysis.

 Detailed suggestions

The title should be changed. In the manuscript, the characterization of cold-pressed oil is just an excerpt.

Keywords: „characterization” of what? of oils, of safflower seeds, etc.

Introduction

The introduction should be expanded. Currently, it is very limited.

The authors should rethink this part in light of the results obtained. One suggestion is to expand the introduction to include characteristics of the raw material and the obtained oil: acid profile, tocopherols, sterols and bioactive compounds (phenolic and others). This will also help in the discussion of the results. However, this is only one suggestion. The authors should deepen their literature studies on the topic.

Line 29-30. Why do the authors claim that the traditional methods of oil pressing are hot pressing and extraction with an organic solvent? Rather, the primary methods of oil extraction are cold pressing.

Line 40-41. Why do the authors believe that the efficiency of cold pressing is low? She is lower than hot pressing or extraction but, in general, in most cases for cold-pressed oils you can get an efficiency of 60-80%. This is due to the fact that the process of conditioning the raw material before pressing is used, of course, within the temperature range allowed for cold pressing.

Materials and Methods

Standards methods given in the manuscript should have their places in references - in full name and source of origin (line 55-56, 65-67).

 2.3 Preparation of WSO. Was the seed conditioned?

2.4 Determination of physicochemical properties of WSO. Why didn't the authors determine the stability and oxidative quality of the oils, e.g. by determining the peroxide value?

2.5 Fatty acid composition determination of WSO. Was the fatty acid profile determined by an author’s developed method or was it according to a standard method? If according to the standard method, this method should be indicated. The authors' publications should not be autocited here.

2.8 Statistical analysis

The authors did not show any statistical tests comparing the results. The significance of the results? Anova analysis. Etc.

 Results and Discussion

The discussion does not exist in this work. The authors do not even relate their results to those of other researchers. There is no attempt to explain or seek explanations for the differences and phenomena that occurred.

On the basis of which statistical tests the authors write that there were or were not statistical differences between the results (this is repeated many times). In the methodology section, there is no mention of statistical tests. In the tables showing the results, there is also no such information. The authors cannot, based on the result, make such a claim.

Table

The titles of the tables should develop the abbreviations that appear there. A table taken out of the context of the manuscript will be unintelligible. 

Table 4: The sum of tocopherols should also be there. This would make it easier to follow the conclusions

 Conclusions

Conclusions should be rewritten after the work has been revised and expanded.

Minor editing of English language required

Author Response

Response to Reviewer 1’ Comments

The publication Characterization of Cold-pressed Wei Safflower Seed Oil is interesting to some extent and could bring new information regarding the possibility of using cold-pressing as a method of obtaining safflower oil. However, the authors gave a very cursory treatment of the subject from the introduction to the paper, through the planning of the experiment to the discussion of their own results. The paper is short, and full of gaps. There are no statistical tests to describe the results, no discussion, and no in-depth literature analysis.

Detailed suggestions

Question 1: The title should be changed. In the manuscript, the characterization of cold-pressed oil is just an excerpt.

Response 1: Thank you for your suggestion! We have changed the title, “Characterization of Wei Safflower Seed Oil Using Cold-pressing and Solvent Extraction”, Thank you! 

Question 2: Keywords: „characterization” of what? of oils, of safflower seeds, etc.

Response 2: Thank you for your suggestion! We have changed “characterization” to “oil characterization”. Thank you!

Question 3: Introduction

The introduction should be expanded. Currently, it is very limited.

The authors should rethink this part in light of the results obtained. One suggestion is to expand the introduction to include characteristics of the raw material and the obtained oil: acid profile, tocopherols, sterols and bioactive compounds (phenolic and others). This will also help in the discussion of the results. However, this is only one suggestion. The authors should deepen their literature studies on the topic.

Response 3: Thank you for your suggestion! We have added more information in this section. Thank you!

Question 4: Line 29-30. Why do the authors claim that the traditional methods of oil pressing are hot pressing and extraction with an organic solvent? Rather, the primary methods of oil extraction are cold pressing.

Response 4: Thank you for you suggestion! The hot pressing and organic solvent extraction methods are the methods commonly used in industry at present. Thank you!

Question 5: Line 40-41. Why do the authors believe that the efficiency of cold pressing is low? She is lower than hot pressing or extraction but, in general, in most cases for cold-pressed oils you can get an efficiency of 60-80%. This is due to the fact that the process of conditioning the raw material before pressing is used, of course, within the temperature range allowed for cold pressing.

Response 5: Thank you for your comment! The cold pressing method is not conducive to continuous production, and under the current equipment conditions, it is difficult to complete the extraction of more than 80% of the oil in the raw material at one time. Thank you!

Question 7: Materials and Methods

Standards methods given in the manuscript should have their places in references - in full name and source of origin (line 55-56, 65-67).

Response 7: Thank you for your comment! These methods are commonly used for oil characterization, and the experimental steps are standardized. Detailed explanations of these methods will cause the length ratio of the article to be out of proportion and a large number of duplicate contents. Therefore, we briefly introduce these methods in this revised manuscript. Many articles also describe these methods in this way.  Thank for you understanding!

Question 8: 2.3 Preparation of WSO. Was the seed conditioned?

Response 8: Thank you for your suggestion! High moisture content is not conducive to oil extraction, so the seed was not conditioned. Thank you!

Question 9: 2.4 Determination of physicochemical properties of WSO. Why didn't the authors determine the stability and oxidative quality of the oils, e.g. by determining the peroxide value?

Response 9: Thank you for your suggestion! In this study, we use Rancimat test to evaluated the stability and oxidative quality, which can comprehensively reflect the quality of oil. Thank you!  

Question 10: 2.5 Fatty acid composition determination of WSO. Was the fatty acid profile determined by an author’s developed method or was it according to a standard method? If according to the standard method, this method should be indicated. The authors' publications should not be autocited here.

Response 10: Thank you for your suggestion! This measurement was done using standard methods, but the details of the experiment may vary due to differences in instrumentation, so we cited our previous study. Thank you for your understanding!

Question 11: 2.8 Statistical analysis

The authors did not show any statistical tests comparing the results. The significance of the results? Anova analysis. Etc.

Response 11: Thank you for your suggestion! The statistical analysis has been added in the tables. Thank you!

Question 12:  Results and Discussion

The discussion does not exist in this work. The authors do not even relate their results to those of other researchers. There is no attempt to explain or seek explanations for the differences and phenomena that occurred.

Response 12: Thank you for your understanding! We have cited many previous reports and added more discussion in section 3.2-3.5. Thank you!

Question 13:  On the basis of which statistical tests the authors write that there were or were not statistical differences between the results (this is repeated many times). In the methodology section, there is no mention of statistical tests. In the tables showing the results, there is also no such information. The authors cannot, based on the result, make such a claim.

Response 13: Thank you for your suggestion! The statistical analysis has been added in the tables. Thank you!

Question 14: Table

The titles of the tables should develop the abbreviations that appear there. A table taken out of the context of the manuscript will be unintelligible. 

Response 14: Thank you for your suggestion! The full name of CP-WSO and HE-WSO were provided in the tables. Thank you!

Question 15: Table 4: The sum of tocopherols should also be there. This would make it easier to follow the conclusions

Response 15: Thank you for your suggestion! The sum of tocopherols was added. Thank you!

Question 16:  Conclusions

 Conclusions should be rewritten after the work has been revised and expanded.

Response 16: Thank you for your suggestion! We revised the conclusion section! Thank you!

Reviewer 2 Report

Comments to the Author,

I consider the manuscript "Characterization of Cold-pressed Wei Safflower Seed Oil” is interesting. However, as I explain in the corrections and observations made, there are some aspects should be consider in order enhancing the research.

Here are some comments for the authors:

In the abstract section the authors say “Our results provides a reference for the development of Wei safflower seeds”. Why? and the verb is provide.

In the introduction section. It is suggested to check the following information “cold pressing ≤ 70 °C”. CODEX STANDARD FOR EDIBLE FATS AND OILS NOT REGULATED BY INDIVIDUAL STANDARDS CODEX STAN 19-1981 (Rev. 2-1999) establishes that “Cold-pressed fats and oils are understood to be vegetable fats and oils edibles obtained, without modifying the oil, by mechanical procedures, for example, extrusion or pressing, without the application of heat. They may have been purified by washing, sedimentation, filtration and centrifugation only”.

In the materials and methods section, it is suggested to specify the temperature under which the extraction processes: pressing and n-hexane were carried out. Room temperature? 25, 30, 35°C?

Table 1. The composition of Weihui safflower seeds, is it expressed in dry or wet basis?

Lines 104 to 105. “In this experiment, the yield of CP-WSO (9.26%) was significantly inferior to that of HE-WSO (16.42%)”. How are the extraction yields expressed, based on the amount of oil available?

Lines 107 to 109 “However, the moisture and volatile substances of CP-WSO were significantly lower than that of HE-WSO, which could be due to the introduction of a small amount of water into the oil during the solvent extraction process”, why was water incorporated during solvent extraction? Improve the explanation from a technological approach.

In Table 2. Oil yield (%, units?), Density and Refractive index (temperature?), Acid value (mg/g, mg of KOH?), volatile substances (%, units?)

It is suggested to incorporate statistical analysis of the results (ANOVA).

Table 4. Tocopherol contents of HE-WSO and CP-WSO (mg/100 g, of oil?)

Lines 137 to 139. “In Table 5, the induction period of CP-WSO was significantly higher than that of HE-WSO….” How do authors know the difference is statistically significant between the induction periods? On the other hand, this idea seems to be contradictory: “…indicating that the stability of CP-WSO was obviously superior to that of HE-WSO, which may be attributed to the higher vitamin E content of HE-WSO”.

The conclusions must be improve.

Although I am not qualified to evaluate the quality of the English in this document, I do detect errors in expression and tenses that should be saved.

Author Response

Response to Reviewer 2’ Comments

I consider the manuscript "Characterization of Cold-pressed Wei Safflower Seed Oil” is interesting. However, as I explain in the corrections and observations made, there are some aspects should be consider in order enhancing the research.

Here are some comments for the authors:

Question 1: In the abstract section the authors say “Our results provides a reference for the development of Wei safflower seeds”. Why? and the verb is provide.

Response 1: Thank you for your suggestion! We have revised this expression, “Our results provide a reference for the development of Wei safflower seed oil”. Thank you!

Question 2: In the introduction section. It is suggested to check the following information “cold pressing ≤ 70 °C”. CODEX STANDARD FOR EDIBLE FATS AND OILS NOT REGULATED BY INDIVIDUAL STANDARDS CODEX STAN 19-1981 (Rev. 2-1999) establishes that “Cold-pressed fats and oils are understood to be vegetable fats and oils edibles obtained, without modifying the oil, by mechanical procedures, for example, extrusion or pressing, without the application of heat. They may have been purified by washing, sedimentation, filtration and centrifugation only”.

Response 2: Thank you for your suggestion! We have revised it according to your comment, “ Oil production by cold pressing refers to the process of destroying oil plant cells by mechanical procedures without the application of heat”. Thank you!

Question 3: In the materials and methods section, it is suggested to specify the temperature under which the extraction processes: pressing and n-hexane were carried out. Room temperature? 25, 30, 35°C?

Response 3: Thank you for your suggestion! We have revised it according to your comment, “The crushed seeds were soaked in 10 times volume of n-hexane at 25 °C for 3 h”. Thank you!

Question 4: Table 1. The composition of Weihui safflower seeds, is it expressed in dry or wet basis? The composition of Weihui safflower seeds, is it expressed in dry or wet basis?

Response 4: Thank you for your suggestion! The composition in Table 1 was expressed in wet basis. The moisture of the seeds was 6.04%, which was also provided in Table 1. Thank you!

Question 5: Lines 104 to 105. “In this experiment, the yield of CP-WSO (9.26%) was significantly inferior to that of HE-WSO (16.42%)”. How are the extraction yields expressed, based on the amount of oil available?

Response 5: Thank you for your suggestion! We have revised section 2.3 to provide the information about the oil yield. Thank you!

Question 6: Lines 107 to 109 “However, the moisture and volatile substances of CP-WSO were significantly lower than that of HE-WSO, which could be due to the introduction of a small amount of water into the oil during the solvent extraction process”, why was water incorporated during solvent extraction? Improve the explanation from a technological approach.

Response 6: Thank you for your suggestion! We have revised this section. The moisture and volatile substances of CP-WSO were significantly lower than that of HE-WSO, which could be due to the residue of the solvent used in the extraction process. Thank you!

Question 7: In Table 2. Oil yield (%, units?), Density and Refractive index (temperature?), Acid value (mg/g, mg of KOH?), volatile substances (%, units?)

Response 7: Thank you for your suggestion! We have added the corresponding units. Thank you!

Question 8:It is suggested to incorporate statistical analysis of the results (ANOVA).

Response 8: Thank you for your suggestion! We have added the ANOVA analysis in the tables. Thank you!

Question 9: Table 4. Tocopherol contents of HE-WSO and CP-WSO (mg/100 g, of oil?)

Response 9: Thank you for your suggestion! We have revised it, “mg/100 g oil”. Thank you!

Question 10: Lines 137 to 139. “In Table 5, the induction period of CP-WSO was significantly higher than that of HE-WSO….” How do authors know the difference is statistically significant between the induction periods? On the other hand, this idea seems to be contradictory: “…indicating that the stability of CP-WSO was obviously superior to that of HE-WSO, which may be attributed to the higher vitamin E content of HE-WSO”.

Response 10: Thank you for your suggestion! The long induction time indicated the good stability. The induction time was calculated automatically by Rancimat instrument. We have added the corresponding information in section 2.7, and figure 1 about Rancimat analysis. Thank you!  

Question 11:The conclusions must be improve.

Response 11: Thank you for your suggestion! We have revised the conclusion! Thank you!

Question 12:Although I am not qualified to evaluate the quality of the English in this document, I do detect errors in expression and tenses that should be saved.

Response 12: Thank you for your suggestion! We have carefully checked and revised the language of this manuscript. Thank you!

Reviewer 3 Report

Comments:

-It should be noted that it is a comparative study by adding the solvent extraction system to the main title.

-If Wei is a safflower variety, it should be specified as "Wei" in the article. If Wei is a location, this formmay not be written.

--The summary part should contain more result figures

-In the introduction, the topics should be given in separate paragraphs according to the similarity. Although cold press is used, no explanatory information about cold press is given.

-The purpose of the study should be stated more clearly. It should not be simple.

- In the material part, detailed information should be given about Safflower rather than chemical reagents. Did you raise it yourself, where was it obtained? In what year? At what stage is the harvest time?

- 2.2. Reference should be given for analyzes in 2.2.chemical composition determination and 2.4. Determination of physicochemical properties of WSO. just analytical code is not enough.

- It should be clearly stated that the crushed seed obtained from the press was used in the extraction with n-hexane. if this shape was made, the oil content result may not be realistic. because during the press, there may be a significant oil loss due to contaminations to the press.

- Agilent 7890A's place of manufacture and company name must be given

- Why was analysis of variance not applied to the results?

- In general, the reason for the high fatty acid composition of cold press oils should be explained.

- The relationship between rancimat values and fatty acids should be explained.

- What is the reason why beta tocopherol and beta tocotrienol are found in cold press oil but not in oil obtained by hexane extraction? There may be an analytical error here.

- Oxidation stability and induction rate should be specified  and discussed in the Ransimat test.

- The conclusion part is insufficient and important results should be emphasized.

-The English of the article should be checked thoroughly. There are some typos.

-The following article should be discussed with the results:

- The Effect of Different Solvent Types and Extraction Methods on Oil Yields and Fatty Acid Composition of Safflower Seed. J Oleo Sci 2019 Nov 7;68(11):1099-1104.

- Fatty acid composition and tocopherol profiles of safflower (Carthamus tinctorius L.) seed oils. Nat Prod Res 2015;29(2):193-6.

- Composition and Characteristics of Some Seed Oils. Asian Journal of Chemistry; Vol. 23, No. 4 (2011), 1851-1853  

should be improved

Author Response

Response to Reviewer 3’ Comments

Question 1: -It should be noted that it is a comparative study by adding the solvent extraction system to the main title.

Response 1: Thank you for your suggestion! We have revised the title, “Characterization of Wei Safflower Seed Oil Using Cold-pressing and Solvent Extraction”. Thank you!

Question 2: -If Wei is a safflower variety, it should be specified as "Wei" in the article. If Wei is a location, this formmay not be written.

Response 2: Thank you for your suggestion! The growth environment has a great influence on the quality of safflower. Many characteristics of safflower produced in Weihui are different from those grown in other places, so we still retain "Wei" in the title to highlight the characteristics of this study. Thank you for your understanding!

Question 3: --The summary part should contain more result figures

Response 2: Thank you for your suggestion! We have added the figure about Rancimat results (Figure 1). Thank you!

Question 4:-In the introduction, the topics should be given in separate paragraphs according to the similarity. Although cold press is used, no explanatory information about cold press is given.

Response 4: Thank you for your suggestion! We have added the explanatory information about cold pressing. Thank you!

Question 5:-The purpose of the study should be stated more clearly. It should not be simple.

Response 5: Thank you for your suggestion! We have revised the introduction section to added more information about the purpose of this study. Thank you!

Question 6:- In the material part, detailed information should be given about Safflower rather than chemical reagents. Did you raise it yourself, where was it obtained? In what year? At what stage is the harvest time?

Response 6: Thank you for your suggestion! We have revised the material part to added more information about safflower seeds. Thank you!

Question 7:- 2.2. Reference should be given for analyzes in 2.2.chemical composition determination and 2.4. Determination of physicochemical properties of WSO. just analytical code is not enough.

Response 7: Thank you for your suggestion! We revised section 2.2, provided more information about chemical composition determination.

Question 8:- It should be clearly stated that the crushed seed obtained from the press was used in the extraction with n-hexane. if this shape was made, the oil content result may not be realistic. because during the press, there may be a significant oil loss due to contaminations to the press.

Response 8: Thank you for your suggestion! The crushed seeds were divided into two parts for the production of CP-WSO and HE-WSO. According to your comment, we have revised section 2.3. Thank you!

Question 9:- Agilent 7890A's place of manufacture and company name must be given

Response 9: Thank you for your suggestion! The corresponding information about this instrument has been added. Thank you!

Question 10:- Why was analysis of variance not applied to the results?

Response 10: Thank you for your suggestion! The analysis of variance has been added in the tables. Thank you!

Question 11:- In general, the reason for the high fatty acid composition of cold press oils should be explained.

Response 11: Thank you for your suggestion! Wei safflower seed is rich in unsaturated fatty acids. As a result, CP-WSO and HE-WSO had the high unsaturated fatty acid composition. Thank you!   

Question 12:- The relationship between rancimat values and fatty acids should be explained.

Response 12: Thank you for your suggestion! We have added the information about the relationship between rancimat values and fatty acids in section 3.5 . Thank you!

Question 13:- What is the reason why beta tocopherol and beta tocotrienol are found in cold press oil but not in oil obtained by hexane extraction? There may be an analytical error here.

Response 13: The cold-pressing method destroys the plant tissue through mechanical extrusion, and extrudes the oil from the sample, and the composition of the oil is close to the natural state.The solvent method mainly extracts oil from plants by the similar compatibility principle, which has selectivity in the extraction process.The content of beta-tocopherol in safflower seed is low, when a large amount of alpha-tocopherol exists, n-hexane may not have a good dissolution effect on beta-tocopherol, so the content of beta-tocopherol in the extracted oil is too low, which is not conducive to HPLC detection (Figure 1 and 2). We added the corresponding discussion in section 3.4.

Question 14:- Oxidation stability and induction rate should be specified and discussed in the Ransimat test.

Response 14: Thank you for your suggestion! We have added the information about induction time in section 2.7. The relationship between oxidation stability and induction time was added in section 3.5. Thank you!

Question 15:- The conclusion part is insufficient and important results should be emphasized.

Response 15: Thank you for your suggestion! We have strengthened the conclusion part. Thank you!

Question 16:-The English of the article should be checked thoroughly. There are some typos.

Response 16: Thank you for your suggestion! We have checked and revised the English language of this manuscript. These revisions are marked in red color. Thank you!

Question 17:-The following article should be discussed with the results:

- The Effect of Different Solvent Types and Extraction Methods on Oil Yields and Fatty Acid Composition of Safflower Seed. J Oleo Sci 2019 Nov 7;68(11):1099-1104.

- Fatty acid composition and tocopherol profiles of safflower (Carthamus tinctorius L.) seed oils. Nat Prod Res 2015;29(2):193-6.

- Composition and Characteristics of Some Seed Oils. Asian Journal of Chemistry; Vol. 23, No. 4 (2011), 1851-1853

Response 17: Thank you for your suggestion! We have read and cited these important references, the corresponding discussion was also added. Thank you!

Round 2

Reviewer 1 Report

The authors included only some of the suggested changes, even though in the response the authors wrote that it was added or changed.  The discussion was expanded, but the expansion of the introduction was ignored. 

Minor editing of English language required

Author Response

Thank you for your positive comment! We have revised the introduction section to add more information about cold-pressing. These revisions are marked in red color. Thank you!

Reviewer 3 Report

Author had made corrections  what I suggested

Author Response

Thank you for your positive comment! Thank you!